# Aberrantly Expressed Hsa_circ_0060762 and CSE1L as Potential Peripheral Blood Biomarkers for ALS

**DOI:** 10.3390/biomedicines11051316

**Published:** 2023-04-28

**Authors:** Metka Ravnik Glavač, Massimo Mezzavilla, Ana Dolinar, Blaž Koritnik, Damjan Glavač

**Affiliations:** 1Institute of Biochemistry and Molecular Genetics, Faculty of Medicine, University of Ljubljana, 1000 Ljubljana, Slovenia; 2Department of Biology, University of Padua, 35122 Padua, Italy; massimo.mezzavilla@unipd.it; 3Department of Molecular Genetics, Institute of Pathology, Faculty of Medicine, University of Ljubljana, 1000 Ljubljana, Slovenia; ana.dolinar@mf.uni-lj.si (A.D.); damjan.glavac@mf.uni-lj.si (D.G.); 4Institute of Clinical Neurophysiology, Division of Neurology, University Medical Centre Ljubljana, 1000 Ljubljana, Slovenia; blaz.koritnik@kclj.si; 5Department of Neurology, Faculty of Medicine, University of Ljubljana, 1000 Ljubljana, Slovenia; 6Center for Human Genetics & Pharmacogenomics, Faculty of Medicine, University of Maribor, 2000 Maribor, Slovenia

**Keywords:** amyotrophic lateral sclerosis, ALS, circRNA, biomarker, peripheral blood biomarker, hsa_circ_0060762, CSE1L

## Abstract

Amyotrophic lateral sclerosis (ALS) is a rapidly progressive adult-onset neurodegenerative disease that is often diagnosed with a delay due to initial non-specific symptoms. Therefore, reliable and easy-to-obtain biomarkers are an absolute necessity for earlier and more accurate diagnostics. Circular RNAs (circRNAs) have already been proposed as potential biomarkers for several neurodegenerative diseases. In this study, we further investigated the usefulness of circRNAs as potential biomarkers for ALS. We first performed a microarray analysis of circRNAs on peripheral blood mononuclear cells of a subset of ALS patients and controls. Among the differently expressed circRNA by microarray analysis, we selected only the ones with a host gene that harbors the highest level of conservation and genetic constraints. This selection was based on the hypothesis that genes under selective pressure and genetic constraints could have a major role in determining a trait or disease. Then we performed a linear regression between ALS cases and controls using each circRNA as a predictor variable. With a False Discovery Rate (FDR) threshold of 0.1, only six circRNAs passed the filtering and only one of them remained statistically significant after Bonferroni correction: hsa_circ_0060762 and its host gene CSE1L. Finally, we observed a significant difference in expression levels between larger sets of patients and healthy controls for both hsa_circ_0060762 and CSE1L. CSE1L is a member of the importin β family and mediates inhibition of TDP-43 aggregation; the central pathogenicity in ALS and hsa_circ_0060762 has binding sites for several miRNAs that have been already proposed as biomarkers for ALS. In addition, receiver operating characteristics curve analysis showed diagnostic potential for CSE1L and hsa_circ_0060762. Hsa_circ_0060762 and CSE1L thus represent novel potential peripheral blood biomarkers and therapeutic targets for ALS.

## 1. Introduction

Amyotrophic lateral sclerosis (ALS) is a rapidly progressive adult-onset neurodegenerative disorder in which both upper and lower motor neurons are affected [1]. The majority of the patients die within 3–5 years of the first symptoms [2]. Current clinical diagnosis is based on clinical examination (El-Escorial criteria) [3] and neurophysiological examination (Awaji criteria) [4], while recently the ALS diagnostic index was also described [5]. For researchers and clinicians, identifying and diagnosing ALS is still a significant challenge. Although genetic testing for the most common mutations, such as C9ORF72, SOD1, TDP43, FUS and TBK1, can help with early diagnosis, mutations in these genes account for only a small percentage of all ALS cases, and establishing the correct diagnosis can still take one year or more [6]. As a result, identifying and validating biomarkers that would shorten the diagnostic process and improve diagnostic accuracy is essential [7]. ALS biomarkers are also important in prognosis because they can detect the onset of the disease before clinical symptoms reveal [8]. Indeed, for example, long before symptom onset, significant biological changes, such as structural degeneration of the brain and spinal cord, have been verified in both SOD1 and C9ORF72 mutation carriers [9,10]. There is a large amount of research data on the field of ALS biomarkers, yet few have in fact been validated. The most promising candidates are neurofilament proteins or so-called neurofilaments. Two of them, namely neurofilament light chain (NfL) and a phosphorylated form of neurofilament heavy chain (pNfH) have been validated. Blood NfL levels appear to be elevated in ALS patients and to remain stable over time, making them potentially useful as diagnostic and prognostic markers; however, they are not limited to ALS and have also been linked to other diseases [11]. Therefore, reliable and easy-to-obtain biomarkers are an absolute necessity for earlier and more accurate ALS diagnoses. Our research group previously described the potential use of circular RNA (circRNA) expression levels in ALS patients as biomarkers [12]. CircRNAs represent a class of non-coding RNAs that are formed during precursor mRNA processing via back-splicing events. Because they do not have 5′ and 3′ ends, circRNAs are resistant to RNA exonucleases and are thus very stable in cells [13]. They have several biologically diverse functions, such as miRNA sponges, protein-coding RNAs and transcriptional regulators [14]. They have already been associated with numerous diseases of the nervous system, such as Parkinson’s disease [15,16], Alzheimer’s disease [17,18], glioblastoma [19,20], multiple sclerosis [21] and epilepsy [22].

A previous work [23] demonstrated the importance of highlighting genes and critical genomic regions subjected to strong purifying selection, showing that such regions are enriched for disease-causing variants and thus should be prioritized in genetic studies that aim to find the causal actors for a disease. These gene prioritization approaches will help identify the parts of the human genome increasingly likely to influence the risk of disease [24].

Here, we wanted to use an approach based on metrics of genetic constraints for the selection of potential disease-causing circRNAs biomarkers. We focused on circRNA host genes that are under selective pressure and have strong genetic constraints as these genes could have a predominant role in determining the disease.

In our work we provided selection of circRNAs for the analysis based on the strength of the genetic constraints in each host gene, relative expression fold change for each circRNA and, finally, function of the host gene. A statistically significant difference (after Bonferroni correction) in expression between patients and controls was observed for hsa_circRNA_060762 and its host gene CSE1L. This approach showed great potential for use as blood-based biomarkers for ALS and for further elucidation of pathologic mechanisms.

## 2. Materials and Methods

### 2.1. Samples

Patients were diagnosed with ALS at the Institute of Clinical Neurophysiology, University Medical Centre Ljubljana, Slovenia. Sixty patients (30 females and 30 males) and 25 age- and sex-matched unrelated healthy Slovenian blood donors without any systemic disease as controls were included in the study. Detailed clinical characteristics of patients are shown in Table 1.

### 2.2. RNA Extraction

Peripheral blood mononuclear cells (PBMCs) were isolated from fresh blood. Ficoll density centrifugation (GE Healthcare, Danderyd, Sweden) was used to collect the cells that were afterwards stored at −80 °C in Qiazol reagent (Qiagen, Hilden, Germany). Total RNA was extracted from stored cells using miRNeasy Mini Kit (Qiagen, Hilden, Germany), according to the manufacturer’s instructions. The concentration and purity of total RNA were measured with NanoDrop ND-1000 (ThermoFisher, Waltham, MA, USA).

### 2.3. Microarray Analysis

Microarray analysis of circRNA expression was performed on a subset of 20 samples —12 patients (6 females, 6 males) and 8 age- and sex-matched controls. Samples were prepared and processed as previously described [12]. Briefly, each sample’s total RNA was prepared for microarray analysis, according to the manufacturer’s procedure (Arraystar, Rockville, MD, USA). To enrich circular RNAs, whole RNA was digested with RNase R (Epicentre, Inc., Lindenhurst, IL, USA). Arraystar Super RNA Labeling Kit (Arraystar, Rockville, MD, USA) was used to amplify and transcribe enriched circular RNAs into fluorescent complementary RNA, which was subsequently hybridized onto the Arraystar Human circRNA Array V2 (8 × 15 K, Arraystar, Rockville, MD, USA). After washing the slides, the Agilent Scanner G2505C was used to scan the arrays.

The Agilent Feature Extraction software (version 11.0.1.1) was used to examine the acquired array pictures. The R software limma package was used to do quantile normalization and further data processing. Through volcano plot filtering, statistically significant differences in circRNA expression between two groups were detected. Fold change filtering was performed to identify circRNAs that were differently expressed between two samples. Hierarchical clustering was used to identify distinct circRNA expression patterns among samples.

### 2.4. QPCR

Total RNA was reverse transcribed to cDNA using SuperScript VILO Master Mix (ThermoFisher, USA). Expression levels circRNAs were measured by real-time quantitative PCR (qPCR) using Sybr Select Master Mix (ThermoFisher, Waltham, MA, USA) on the Rotor Gene Q 5 plex HRM platform (Qiagen, Hilden, Germany) in duplicate for each sample. Used primers are shown in Appendix A. Primers were synthesized by IDT (Coralville, IA, USA) or Qiagen (Hilden, Germany) (QuantiTect primers). *RPS17* and *RPL13A* were used as reference genes, and the data were analyzed using the comparative cycle threshold method (2^ΔΔCt^).

### 2.5. Statistical Analysis

Among the differently expressed circRNA by microarray analysis, we collected only the ones with a host gene that harbors any evidence of genetic constraints. This selection was based under the hypothesis that genes under selective pressure and genetic constraints could have a major role in determining a trait or disease. In particular, we collected the probability of loss of function intolerance score (pLI) from gnomAD [25], where genes under constraints are highly intolerant to loss of function variants, which results in a high pLI (pLI > 0.9), and the DSC and SSC score for Europeans [26]. The latter two metrics are population-based, and a negative score in a gene highlights a high-level of constraints to genic variation in both coding and non-coding regions. More in detail, considering that the majority of our patients have European ancestry, we analyzed the genes that showed evidence of purifying selection in Europeans. Then we created a “conserved” set of genes that follow these criteria: pLI = 1 (highest probability of loss of function intolerance), DSC score < −2 and SSC score < −2 [26]. These stringent criteria aim to select the genes with the highest level of genetic constraints and evidence of ongoing purifying selection. Linear regression analyses were performed using the expression level as a predictor and the status (cases and controls) as the response. For regression analyses, we used only the set of genes labeled as “conserved set”. False discovery rate and Bonferroni correction on regression *p*-values were calculated using R [27].

All experimental data were analyzed using SPSS software 24.0 (SPSS, Chicago, IL, USA). Differences in expression levels between patients and healthy controls were assessed using the Mann-Whitney U test. Spearman’s rank correlation was used to determine the correlations between circRNA/gene expression levels and clinical data. The diagnostic potential of circRNA and host genes was assessed with ROC curve analysis. A *p*-value < 0.05 and AUC metric > 0.5 was considered to be statistically significant.

## 3. Results

The number of genes with pLI = 1 and DSC score and SSC score ≤ −2 was 29 from an initial set of 4168 genes. Then we performed the regression analyses using the circRNA mapped on these genes (a total of 95 from an initial set of 10,161). Then, after Bonferroni correction, the only circRNA that was significantly associated (*p*-value ≤ 0.05/95) with ALS status was hsa_circRNA_060762, which is encoded in the CSE1L gene (see Figure 1). Afterwards, we also estimated the false discovery rate (FDR) for each circRNA analyzed among the “conserved set” of genes, using, in this case, a less stringent cut-off of false discovery rate of 0.1. We found that the following genes are associated with ALS status: UPF2, XPOI, KPNB1 and MED13, in addition to CSE1L. Despite not being statistically significant (after Bonferroni correction), we can consider these genes as possible candidates for future analyses of gene–gene interaction (Appendix A).

## 4. Expression of circRNA and Host Gene

The only circRNA that remained significant after Bonferroni correction was hsa_circRNA_060762. We determined the expression levels of hsa_circ_0060762 and its host gene CSE1L (Figure 2). Both circRNA and its host gene were significantly downregulated in 60 ALS patients compared to healthy controls, expression of hsa_circ_0060762 by 2.5-fold and expression of CSE1L by 1.8-fold.

## 5. Associations between Clinical Variables and circRNA Expression

Using the Spearman rank correlation test, we observed no statistically significant association between circRNA expression and clinical parameters. There was a slight positive correlation between the expression of circRNA and its host gene (Appendix A). The main reason that we found no statistically significant association between circRNA expression and clinical parameters could be due to the patient’s sample size, the estimated correlation between circRNA expression and the ALS onset of 0.194 or, with survival time, 0.162. However, the smallest amount of correlation that resulted significantly in our analyses was between ALS onset and disease duration with a magnitude of −0.29. Furthermore, the absence of association might be owing to probable interactions between sex, age and circRNA expression that are not accounted for in the Spearman correlation analysis.

## 6. Diagnostic Potential

We performed receiver operating characteristics (ROC) curve analysis to evaluate the diagnostic potential of hsa_circ_0060762 and CSE1L. The curves for circRNA and its host gene are similar, resulting in an area under the curve (AUC metric) of approximately 0.75, together with 82.5% sensitivity and 62.5% specificity for the optimal cut-off point (Figure 3).

## 7. Discussion

Amyotrophic lateral sclerosis (ALS) is a rapidly progressing neurodegenerative disease that is often diagnosed with a delay due to initial non-specific symptoms. Several types of biomarkers were already proposed (miRNAs, mRNAs, proteins, various metabolites) in cerebrospinal fluid, leukocytes, serum and plasma [28]. None of them is routinely used in the diagnostic at the moment although some show great potential for further validation. CircRNAs have already been proposed as potential biomarkers for several diseases, including Alzheimer’s disease [29], multiple sclerosis [21], major depressive disorder [30], numerous cancers [31] and ALS itself [12,32].

Here, we further investigated the usefulness of circRNAs as potential biomarkers for ALS. Our framework was based on a reproducible gene prioritization approach in order to select the circRNAs with host genes that showed the highest level of conservation and genetic constraints. Because, according to our hypothesis, these conserved genes should be the ones in which any kind of variation should have a more substantial effect on the disease compared to the non-conserved type of genes. Then we performed a linear regression between ALS cases and controls using each circRNA as a predictor variable. With an FDR threshold of 0.1, only six circRNAs passed the filtering, and merely one of them remained statistically significant after Bonferroni correction: hsa_circ_0060762 and its host gene CSE1L. Finally, we observed the significant difference in expression levels between patients and healthy controls for both hsa_circ_0060762 and CSE1L. hsa_circ_0060762 is encoded in the CSE1L gene, which is found in the “conserved set” of ALS genes with the highest level of genetic constraints. CSE1L in humans encodes an exporting factor for importin α. CSE1L is a member of the importin β family and mediates the re-export of importin α proteins to the cytosol. In the spinal cord of the mice model of ALS, an altered localization of two proteins of the nucleocytoplasmic transport system, importin α and importin β, was detected using immunohistochemistry [33]. An abnormal transporter protein distribution was also detected in spinal cords of patients with a sporadic and familial form of ALS [34]. Furthermore, reduced levels of CSE1L were reported in the post-mortem brains of patients with frontotemporal lobar degeneration (FTLD), the disease which shares many clinical, pathological and genetic characteristics with ALS, including nuclear trafficking impairment as well as in the post-mortem brains of ALS-TDP patients [35,36,37]. TDP-43 is translocated into the nucleus via the classical nuclear import pathway involving importin α/β proteins. Inhibition of the importin α/β pathway caused cytosolic retention of TDP-43 [38,39]. TDP-43 can become mislocalized to the cytosol when their respective nuclear import pathway is impaired.

TDP-43 dysfunctions and cytoplasmic aggregation seem to be the central pathogenicity in ALS and FTLD. The dysregulation of RNA metabolism and enhanced cytoplasmic TDP-43 mislocalization are caused by the impairments of TDP-43 autoregulation and nucleocytoplasmic shuttling. Multiple cytotoxic effects, including abnormal stress granule dynamics, liquid–liquid phase separation, mitochondrial dysfunction, endoplasmic reticulum (ER) stress, impaired axonal transport and proteolysis dysfunction, are brought on by the nuclear depletion and cytoplasmic aggregation of TDP-43. TDP-43 aggregates exhibit prion-like cell-to-cell spread, which may contribute to the development of the disease [40]. In addition, recent discoveries in the biology of TDP-43 demonstrate that the importin α/β heterodimer, which functions as both a nucleus import transporter and a cytoplasmic chaperone, inhibits TDP-43 aggregation [41]. The importin α/β pathway seems to be specifically impaired in patients with TDP-43 pathology [42]. Therefore, unraveling both the physiological and pathological mechanisms of TDP-43 may enable the exploration of novel diagnostic and therapeutic approaches [40]. The deregulated circRNA hsa_circ_0060762 has potential physiological functions on its own. CircRNAs that act as miRNA sponges can indirectly control the expression of several genes. During miRNA-sponging, each circRNA interacts with numerous miRNAs to reduce their mRNA silencing potential [43]. Human circRNA Database (circBank) data show that hsa_circ_0060762 (circBank ID: hsa_circCSE1L_066) is 645 nucleotides long and contains binding sites for multiple miRNAs (circBank). Several of these miRNAs have already been linked to ALS because they are differentially expressed in the tissues and body fluids of ALS patients and thus suggested as possible ALS biomarkers. Among them are has-miR-9 [44,45], hsa-miR-125a/b [46,47], hsa-miR-153 [46], has-miR-192 [48], has-miR-371b [49], has-miR-1275 [50] and has-miR-4695 [46]. For example hsa-miR-125b regulates microglia activation and motor neuron death in ALS through IL-6 and STAT3 pathway, respectively, causing an increase in tumor necrosis factor-alpha (TNFα) expression [51]. Another study shows that human neurofilament (NEFL) 3′UTR has two sites for hsa-miR-9 binding and that hsa-miR-9 is capable of reducing the expression of *NEFL* and regulating the mRNA stability of essential intermediate filaments and thus potentially contributing to the pathogenesis of intermediate neurofilament inclusions in ALS [52].

In this study, for the first time, we report the reduced expression of has_circ_0060762 and its host gene CSE1L in peripheral blood mononuclear cells of patients with ALS compared to controls. Our approach combined both genetic prioritization based on in silico predictions of genetic constrains and purifying selection and expression data. This reproducible approach showed great potential for use as blood-based biomarkers for ALS and for further elucidation of pathologic mechanisms. In addition, receiver operating characteristics curve analysis showed some diagnostic potential for CSE1L and hsa_circ_0060762, which is similar to that reported for pNfH [53]. However, a direct comparison between these AUC metrics for circRNA and NfL is impossible because the samples used are different. A future effort should be focused on obtaining both RNA, NfL and pNfH for the patients and estimating the single and combined sensitivity and specificity. hsa_circ_0060762 and/or CSE1L thus represent a novel potential biomarker for ALS as well as open possibilities for further investigations of their role in ALS pathogenesis and therapy. Before drawing any conclusions about their applicability as possible ALS biomarkers, they need to be validated and compared to other neurodegenerative diseases and the mice model of ALS. Also, some limitations in this study like sample size and reduced power to detect circRNA with smaller effect have to be considered.

In conclusion, we showed that hsa_circ_0060762 and/or CSE1L have the potential to be effective blood-based circulating ALS disease biomarkers. However, an extensive validation based on diverse sets of healthy and diseased cases, preferably with a larger number of samples in each group, has to be performed, before we can classify them as useful biomarkers for ALS.

## Figures and Tables

**Figure 1 biomedicines-11-01316-f001:**
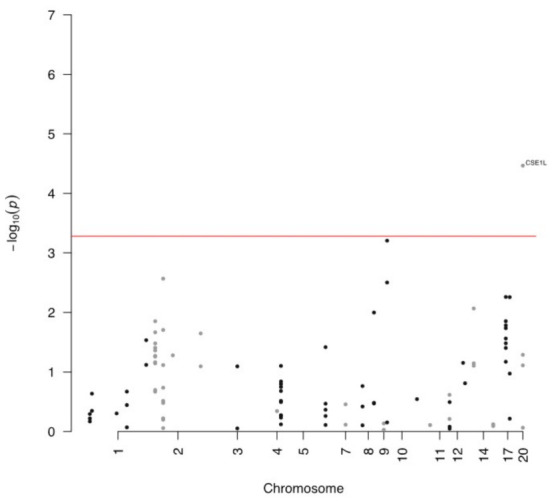
Manhattan plot of circRNA inside the “conserved set” of genes; the red line represents the significance threshold after Bonferroni correction.

**Figure 2 biomedicines-11-01316-f002:**
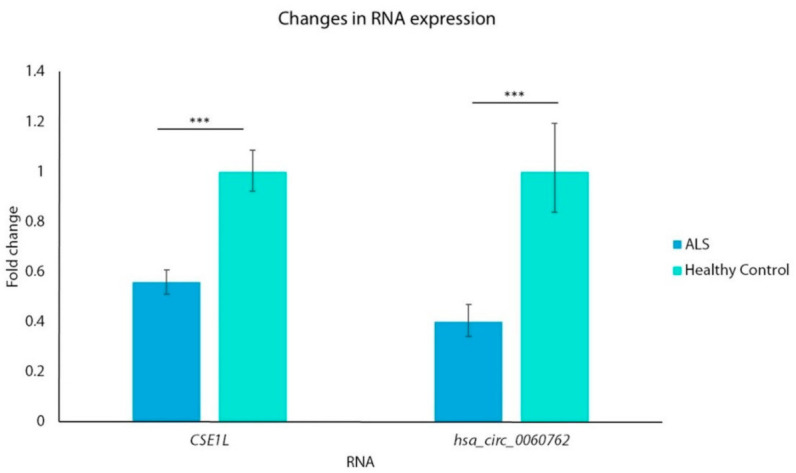
Expression levels of hsa_circ_0060762 and CSE1L. *** *p* < 0.001.

**Figure 3 biomedicines-11-01316-f003:**
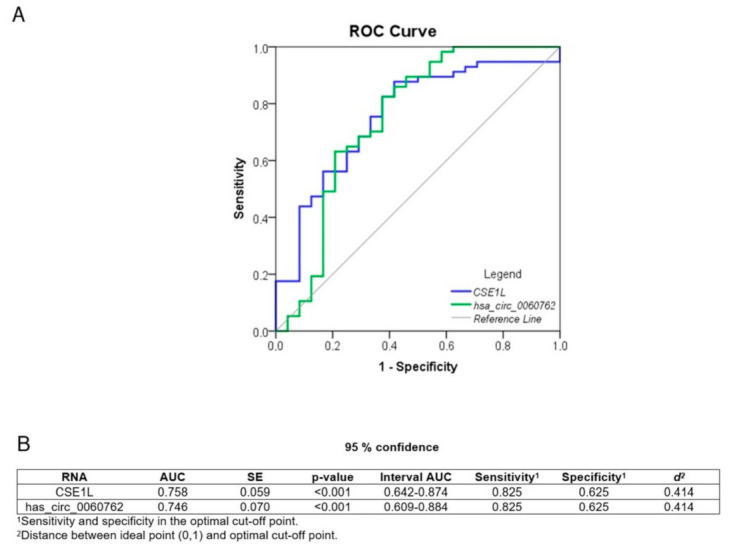
Diagnostic potential of hsa_circ_0060762 and CSE1L. (**A**) ROC curves. (**B**) Detailed information of shown ROC curves.

**Table 1 biomedicines-11-01316-t001:** Detailed clinical characteristics of included patients and healthy controls.

	Samples	
Characteristics	ALS (*n* = 60)	Healthy Controls (*n* = 25)
Sex (M/F)	30/30	15/10
Age (years) ^a^	67 (35–92)	56 (47–73)
Age at onset (years)	65 (35–92)	/
ALS onset (spinal/bulbar/mixed)	45/13/2	/
Disease duration (years) ^b^	1.5 (0.0–5.5)	/
Survival time (years) ^c^	2.0 (0.5–5.0) *n* = 27	/
Level of functional impairment ^d^	34 (20–48)	/
Rate of progression ^e^	−1.11 (−0.03–−4.19)	/

^a^ Age at the time of blood collection. ^b^ Time from symptom onset to blood collection. ^c^ Time from symptom onset to death. ^d^ ALS-FRS-R (ALS functional rating scale revised) points at the time of blood collection. ^e^ Slope of the linear regression line for ALS-FRS-R points.

## Data Availability

The datasets during and/or analyzed during the current study are available from the corresponding author on reasonable request.

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
