# Peer review of "Aberrantly Expressed Hsa_circ_0060762 and CSE1L as Potential Peripheral Blood Biomarkers for ALS"

_biomedicines, 2023, doi:10.3390/biomedicines11051316_

Round 1
Reviewer 1 Report
Review of the study:Aberrantly expressed Hsa_circ_0060762 and CSE1L as potential 2 peripheral blood biomarkers for ALS
Abstract
Therefore, reliable and easy-to-obtain biomarkers are in desperate need for earlier and more accurate diagnostics – wykreślić słowo desperate
Introdaction
-Majority of 43 the patients die within 3 years of first symptoms (2) – ranges are 3-5 years
-Current clinical diagnosis is based 44 on clinical examination (El-Escorial criteria) (3) and neurophysiological examination 45 (Awaji criteria) (4), while recently the ALS diagnostic index was also described (5). For 46 Biomedicines 2022, 10, x FOR PEER REVIEW 2 of 10 researchers and clinicians, identifying and diagnosing ALS is still a significant challenge. 47 Although genetic testing for the most common mutations, such as C9ORF72, SOD1, 48 TDP43, FUS, and TBK1, can help with early diagnosis, mutations in these genes account 49 for only a small percentage of all ALS cases and establishing the correct diagnosis can 50 still take one year or more (6) -I would move this passage to the method and relate it to the group presented
Separate the subsection coclusion
Reviewer 2 Report
Aberrantly expressed Hsa_circ_0060762 and CSE1L as potential peripheral blood biomarkers for ALS
Authors mention that the hsa_circRNA_060762 and its host gene CSE1L are peripheral blood biomarkers for ALS This manuscript has some value to be published in Biomedicines, but there are some specific comments.
Comments to the authors.
Major comments
1) The AUC metric for circRNA and its host gene are approximately 0.75, together with 82.5% sensitivity and 62.5% specificity for the optimal cut-off point (Fig. 3). Are these sensitivity and specificity stronger than the neurofilament light chain (NfL) and phosphorylated form of neurofilament heavy chain (pNfH)?
2) Authors observed no statistically significant association between the circRNA expression and clinical parameters of ALS patients. There is no discussion about this point.
3) How do authors select the 25 age- and sex-matched healthy controls? Is the electrophysiological study performed these “healthy controls”?
4) It would be better to compare the circRNA in mice model of ALS and in other neurodegenerative disorders.
Minor comments:
L210~217: I think that the part of TDP43 is rather bored.
